# In Vivo Efficacy of Purified Quillaja Saponin Extracts in Protecting against *Piscirickettsia salmonis* Infections in Atlantic Salmon (*Salmo salar*)

**DOI:** 10.3390/ani13182845

**Published:** 2023-09-07

**Authors:** Hernán Cortés, Mario Castillo-Ruiz, Hernán Cañon-Jones, Trinidad Schlotterbeck, Ricardo San Martín, Leandro Padilla

**Affiliations:** 1Desert King Chile, Viña del Mar 2420505, Chile; tschlotterbeck@desertking.cl (T.S.); lpadilla@desertking.cl (L.P.); 2Escuela de Química y Farmacia, Facultad de Medicina, Universidad Andres Bello, Santiago 8370134, Chile; mario.castillo.r@uandresbello.edu; 3Departamento de Ciencias Químicas y Biológicas, Facultad de Ciencias de la Salud, Universidad Bernardo O’Higgins, Santiago 8370854, Chile; 4Núcleo de Investigación Aplicada en Ciencias Veterinarias y Agronómicas, Facultad de Medicina Veterinaria y Agronomía, Universidad de Las Américas, Santiago 7500975, Chile; 5Sutardja Center for Entrepreneurship and Technology, College of Engineering, University of California, Berkeley, CA 94720, USA; rsanmartin@berkeley.edu

**Keywords:** piscirickettsiosis, quillaja extract, saponins, salmonids, immunomodulation

## Abstract

**Simple Summary:**

The global population is growing at higher rates and food production must grow at a similar rate. Aquaculture is one the most important sources of protein for human consumption, and its sustainable growth is of vital importance. The Chilean salmon industry exported 751,000 metric tons of fish in 2022; however, many diseases affected the production of these products. *Piscirickettsia salmonis* is the main bacteria infecting salmon in Chile, causing 10% of losses. Currently, no successful treatment exists against this bacterium. Quillaja saponins are secondary metabolites with biological applications, and have shown antiviral, antifungal, and antibacterial properties, as well as acting as immunomodulators. Here, we tested the effect of quillaja extracts in protecting against *P. salmonis* infection. These products were applied as a supplement in the feed, and displayed in different model trials protective effect against the infection. All treated groups showed a higher survival percentage in comparison to the control group. In addition, quillaja extracts were able to increase the immune responses measured throughout different cytokines. Finally, in a pilot-scale trial, it was observed that fish fed with quillaja saponins exhibited a reduction in mortality and a reduction in the need for antibiotic treatment. These results represent evidence that quillaja saponins are a potential natural non-pharmacological strategy to prevent intracellular infections.

**Abstract:**

Piscirickettsiosis, the main infectious disease affecting salmon farming in Chile, still has no efficient control measures. *Piscirickettsia salmonis* is a facultative intracellular bacterium that can survive and replicate within the host macrophages, evading the immune response. Triterpenic saponins obtained from the *Quillaja saponaria* tree have been widely studied, and have been shown to be immunomodulatory agents, suitable for feed and vaccine applications for veterinary and human uses. The impact of the oral administration of two extracts of Quillaja saponins on the infection of *P. salmonis* in *Salmo salar* and the corresponding gene expressions of immunomarkers were studied under three in vivo models. In the intraperitoneal challenge model, the group fed with Quillaja extracts showed lower mortality (29.1% treated vs. 37.5% control). Similar results were obtained in the cohabitation model trial (36.3% vs. 60.0%). In the commercial pilot trial, the results showed a significant reduction of 71.3% in mortality caused by *P. salmonis* (0.51% vs. 1.78%) and antibiotic use (reduction of 66.6% compared to untreated control). Also, Quillaja extracts significantly modulated the expression of IFN-II and CD8. These results represent evidence supporting the future use of purified Quillaja extracts as a natural non-pharmacological strategy for the prevention and control of *P. salmonis* infections in salmon.

## 1. Introduction

Piscirickettsiosis is caused by the etiological agent *Piscirickettsia salmonis*, a Gram-negative facultative intracellular bacterium [1], and the first rickettsia-like bacterium to be known as a fish pathogen [2]. In Chile, *P. salmonis* outbreaks are a major cause of mortality in salmon farming during the growing phase in seawater. During the first part of 2022, *P. salmonis* infections accounted for 54.2% of mortality in Atlantic salmon related to infectious diseases and 10.17% of total mortality in that specie [3]. Moreover, *P. salmonis* infections may be regarded as the main cause behind the increase in antimicrobial use in the Chilean salmon industry. In 2021, Chile used 463,400 kg of antimicrobials (~0.47 kg per ton of harvested salmon), with florfenicol and oxytetracycline being the most frequently used antibiotics to treat *P. salmonis* [4,5,6]. Some studies have suggested that a high percentage of the antibiotics used in the salmon industry ultimately reach the sea, negatively impacting the environment, as well as microbial communities [4].

Commercial vaccines against *P. salmonis* infection are currently available; however, these are unable to confer full protection against this pathogen under field conditions during the seawater life cycle of Chilean salmon. Therefore, several immunizations are normally required to achieve an effective and protective immune response against this pathogen [7]. The intracellular nature of *P. salmonis*, as well as the complex infection strategies employed by this pathogen, can explain the success of its infectious process and the failure of currently available treatments [1,5,8,9]. Therefore, new non-pharmacological strategies against this bacterium must be developed and evaluated for implementation at the commercial scale.

In general, intracellular pathogens such as *P. salmonis* modulate the innate and adaptive immune responses of fish through several virulence factors that promote a favorable intracellular environment suitable for survival, replication, and chronic maintenance in host cells [7]. Different studies have demonstrated that the pathology of *P. salmonis* during active infection in salmon relies on the cytokine imbalance biased toward Th2 type immune response, in addition to the evasion of other immune mechanisms by replication into phagocytes [10].

Proinflammatory cytokines such as IL-1β, IL-8, IL-12 and IFN are some of the main markers induced by the innate immune system at the onset of the infection, and their activity can impact the kind of adaptive immune response that follows that process. In fish, IL-1β is involved in the inflammatory responses against Gram-negative bacteria [11]. IL-8 is a chemokine that regulates the neutrophil activation and transcription of pro-inflammatory cytokines [12]. In addition, the immunodetection and potential antimicrobial activity of IL-8 has been shown in rainbow trout [13]. IL-12 is a key component for the efficient performance of phagocytes in teleost fish. IL-12 stimulates IFN-γ production, promoting the activity of cytotoxic T cells and stimulating T helper 1 (Th1) cells. Type I IFNs (IFN-I) are classified as α or β and are involved in the inhibition of viral replication into infected cells [14]. IFN-II (IFN-II), also referred to as IFN-γ, is associated with protection against intracellular pathogens, and activates macrophages, inducing their killer activity against the pathogens and changing the immune response to Th1 [15,16].

In order to achieve an effective control *P. salmonis* infections, the induction of a combined Th1/Th2 immune response is critical, leading to balanced kinetics of the pro-inflammatory and anti-inflammatory cytokine response. Such a balance between Th1/Th2 is critical to achieving the removal of intracellular pathogens, as well as the equilibrium of the immune system [10]. Therefore, the development of strategies based on the modulation of interleukin production would represent a major milestone in the road toward successful immunological strategies for the reduction in the damage and mortalities generated by the disease in the salmon industry [10].

In that regard, the use of the extracts of *Quillaja saponaria,* Molina (Quillay), could be a promising approach to develop such strategies. *Q. saponaria* is an evergreen tree native to Chile; this plant is a well-known source of extracts rich in immunostimulant saponins of the triterpenic type [17]. These saponins are composed of a hydrophobic core or aglycone, called sapogenin, and two sugar branches; this molecular arrangement confers to these molecules an amphipathic character and particular chemical and biological properties [18]. In addition to saponins, quillaja extracts contain other components such as polysaccharides and phenolics, mostly the aromatic piscidic acid [19]. Although quillaja extracts have been used extensively as emulsifiers for foods and beverages, foaming agents, nematicides in agriculture and in many other applications [18,19], it is well known that some quillaja saponins are able to block viral and bacterial association with cells, as well as modulate intracellular processes. These biological effects lead to the induction of an innate and adaptive immune response, including cellular and humoral types [20]. High-grade quillaja extracts containing ≥90% *w*/*w* saponins (dry basis) can be employed as immunostimulant agents (vaccine adjuvants) in veterinary and human vaccine formulations [20,21], where they effectively modify the activities of antigen-presenting cells (macrophages and dendritic cells). It has been demonstrated that quillaja saponins, alone or incorporated into immune-stimulating complexes (ISCOMs), modulate immunity responses in different ways: by increasing antigen uptake in antigen-presenting cells (APC), by stimulating Th1 and Th2 immune responses to different antigens, and by inducing cytotoxic T lymphocytes (CTL) [22].

A previous report from our research team showed that quillaja saponin extracts were able to reduce *P. salmonis* infections on SKH-1 cell cultures, and the efficacy was dependent on the quantity of the saponins applied to the cells [23]. Specifically, quillaja extracts had the potential to reduce infections by 10- to 1000-fold according to in vitro assays [23]. In addition, purified quillaja saponin extract promotes phagosome–lysosome fusion and a balance in the expression of IL-12 and IL-10 in macrophage cell lines [20].

The aim of this work was to test the effect of the feeding of Atlantic salmon with formulations containing purified quillaja saponin extracts on the mortality resulting from *P. salmonis* infection.

## 2. Materials and Methods

### 2.1. Quillaja Saponin Extracts

Two powder *Q. saponaria* extracts with different levels of purification and saponins were used for the in vivo trials: (a) PAQ-Xtract (highly purified in triterpenic saponins), which corresponds to a quillaja extract type II defined in FAO JECFA; and (b) a non-purified quillaja extract (NPQ) (2.5 times lower level of purified saponins compared to PAQ-Xtract), which corresponds to a quillaja extract type I defined by FAO JECFA (REF). As described by FAO JECFA, quillaja extract type II has a saponin content of at least 65% and no more than 90% in solids, compared to quillaja extract type I, which has a content of between 20 to 26% saponins. The products were sourced from Desert King Chile (Av. Industrial 1970, Quilpué, Chile).

### 2.2. In Vivo Efficacy of Quillaja Saponin Extracts against P. salmonis in Freshwater Conditions under Intraperitoneal Challenge

The first trial was conducted at the ACTIVAQ laboratory in Universidad de Santiago de Chile. One hundred and eighty (180) Atlantic salmon fry weighing 30 ± 1.4 g/each were used. The fish employed had no history of infectious salmon anemia virus (ISAv) or *P. salmonis* infection, as verified via sampling and subsequent analysis of the molecular diagnosis using real-time RT-PCR. Prior to being transferred to the experimental station, 60 fish were sampled to check their health status through gill inspection, intestine and skin sampling, Gram staining in their internal organs (spleen, kidney, and brain), staining with acridine orange in their gills, IFAT analysis for bacterial kidney disease (BKD) and *P. salmonis*, and RT-PCR for infectious pancreatic necrosis virus (IPNv). During the trial, the fish were fed with a diet of 15 micro EWOS 15CP^®^ (50.0% total crude protein, 22% lipids, 1.0% crude fiber, 9.0% moisture, 9.5% ash and 8.5% nitrogen-free extract), at a daily rate of 0.75% of body weight/day (bw/day). Both products were included in the diet at a dose of 3.75 mg bw/day. The optimal dose of quillaja extracts used in the feed in these studies was determined based on previous studies where incremental doses of quillaja extracts where given orally to the fish, evaluating a range from 0.9 to 12.0 mg of saponins/kg body weight of the fish. The following experimental groups were used: (a) a positive control (fish challenged with the bacteria and fed without quillaja extracts products); (b) a negative control (fish not challenged with bacteria and not fed with quillaja extracts products); (c) a PAQ-Xtract group, challenged with the bacteria; and (d) a NPQ group, challenged with the bacteria. All groups were tested in duplicate. The challenge was performed via an intraperitoneal injection of *P. salmonis* genogroup LF-89-like (PS-LF-89) in the ventral line at a rate of 0.1 mL of inoculum (3 × 10^6^ bacterial genome 15 copies/mL) in each fish. Negative control fish were inoculated with 0.1 mL of culture medium. The feeding schedule was carried out with 15 days of acclimation (all fish fed with no quillaja extracts), followed by 45 days of treatment with orally administered quillaja extracts. Infection with *P. salmonis* was carried out at day 7 post-acclimation. Mortality rate was registered daily and calculated at the end of the trial. IFN-II and C3 gene expression was measured from the head kidney and spleen of fish and quantified via RT-qPCR before the challenge with *P. salmonis* and at the end of the trial.

### 2.3. In Vivo Efficacy of Quillaja Saponin Extracts against P. salmonis in Atlantic Salmon Smolts, under a Cohabitation Challenge

The second trial was conducted at the Salmon Clinical Trials Unit in Universidad Austral de Chile, (Valdivia, Chile) using a model of cohabitation challenge in sea water tanks. This trial included a negative control group (no feeding with quillaja extracts and not challenged), a positive control (no feeding with quillaja extracts but challenged), a PAQ-Xtract group with bacterial challenge and an NPQ group with bacterial challenge. All groups were tested in duplicate and distributed randomly in 8 tanks with 30 unvaccinated clinically healthy Atlantic salmon smolts. The initial mean fish weight was 115 g, with fish stocked at 9.64 kg/m^3^. The experimental diets were delivered for a month before starting the co-habitation challenge. Feed was delivered to the fish twice a day, early in the morning and late in the day. The feeding rate was calculated at 2.0% of the fish weight per day for the feeding phase and 1.5% in the challenge phase. Seawater was provided for all tanks at 15–17% salinity, the temperature ranged between 9.6 and 12.6 °C, the oxygen concentration was maintained between 6.2 and 6.8 mg/L, and the photoperiod was set to a 24:0 h light/darkness cycle during the trial. The inoculum consisted of a *P. salmonis* genogroup LF-89-like (PS-LF-89) culture (Biotechnology and Aquatic Pathology Laboratory of Universidad Austral de Chile) and was prepared at 10^5^–10^6^ CFU/mL. The infective fish, called “trojans”, were inoculated intraperitoneally with 0.1 mL of the bacterial suspension under anesthesia (0.5 g/L benzocaine chlorhydrate). Fish weight and length were recorded at the beginning and end of the trial. Mortality was recorded daily for each tank to calculate mortality rates.

### 2.4. RNA Extraction and Quantitative Analyses of Immune Gene Expression by RT-qPCR

Total RNA from the head kidney and spleen was extracted using the E.Z.N.A^®^ RNA extract kit (OMEGA Bio-tek Inc., Norcross, GA, USA) according to the manufacturer’s instructions. The mRNAs were transformed into cDNA using reverse transcription in a two-step procedure. The first step consisted of a reaction using a mixture of 1.6 μL of oligo-dT (1.25 μg/mL), 1.0 μL of dNTPs (10 mM), 8.0 μL of total RNA (5 μg) and 0.1 μL of nuclease-free water and incubated for 10 min at 60 °C to eliminate secondary structures of the mRNAs. Then, the next step was adding to this solution a mixture of 1 μL of M-MLV reverse transcriptase (200 U), 4 μL of 5× enzyme buffer and 0.5 μL of RNAsaOUT recombinant ribonuclease inhibitor (40 U), incubated for 1 h at 37 °C. To inactivate the reverse transcriptase reaction, the mixture was incubated at 72 °C for 10 min. The synthesized cDNA was stored at −20 °C for subsequent PCR amplification or quantification via real-time PCR (qPCR). Each amplification reaction was carried out using 2 μL of cDNA template, 0.2 μM primers, 0.8 μL of MgCl_2_ (25 mM), 1 μL of Lightcycler^®^ Fast Start DNA Master SYBR Green Amplification Mix in a total volume of 10 μL. The tube containing the mixture was placed in a LightCycler^®^1.5 thermocycler; the amplification cycle consisted of the following steps: initial denaturation at 95 °C for 10 min, followed by 35 cycles of denaturation at 95 °C for 10 s, annealing at 58 °C for 10 s, and extension at 70 °C for 10 s. Subsequently, a cycle to obtain the melting curve was run for 20 s at 95 °C, and finally a cooling cycle at 40 °C was run for 30 s. Relative quantification was carried out using a standard curve prepared from reactions containing dilutions of the purified PCR product and a known concentration for the gene of interest. Once the PCR product corresponding to each gene was obtained and quantified, successive dilutions were made in a range of 10^7^ to 10^2^ copy numbers/μL for each gene under study. Subsequent calculation of the reaction efficiency was performed following the relationship of E = 10 (−1/slope) − 1. To calculate the relative expression using the qPCR technique, cDNA amplification reactions of the elongation factor 1 alpha (EF-1) gene were performed in each RNA sample. The expression of EF-1 was suggested as reference gene in RT-qPCR assays for studying the effect of *P. salmonis* on the host immune response [24]. Then, expression changes were calculated using the comparative CT method. The graphs depict the average and standard deviation of the expression fold change calculated using the expression of the control fish as a calibrator. Asterisks depict statistically significant differences according to the unpaired *t* test of (* *p* < 0.05; ** *p* < 0.01; *** *p* < 0.001, **** *p* < 0.0001). The primers used in this study are listed in Table 1.

### 2.5. Efficacy of PAQ-Xtract in the Prevention of P. salmonis Infections in Atlantic Salmon in Seawater under Pilot Commercial Conditions

A trial under pilot commercial conditions during the growing phase in seawater was carried out to test the efficacy of PAQ-Xtract in the prevention and control of naturally occurring *P. salmonis* infections in Atlantic salmon. The trial was conducted at the Huenquillahue seawater facilities of MOWI Chile located 17 km from Puerto Montt (Chile). This facility consists of metallic suspended sea-cages with dimensions of 15 × 15 × 15 m, including a system to continuously measure water quality variables such as temperature and oxygen. The facility was also equipped with an automatic oxygen release device in case of any sudden drop of oxygen, and an automatic feed recovery system. This set up allowed the running of the trial at production scale in a semi-controlled system. The trial included two groups: the control (no PAQ-Xtract added to feedstuff) and treated (with PAQ-Xtract) groups. Each group comprised three cages with 10,000 fish each. Fish were fed with PAQ-Xtract using a strategy of 2 pulses of 2 months each with a 1-month rest interval. Biomass, mortalities, economic and biological feed conversion rate, and the use of antibiotics were recorded during the trial. Antibiotic treatment was measured as the percentage of cages that required the use of antibiotics at least once during the trial.

### 2.6. Statistical Analyses

Statistical differences were evaluated using Student’s *t*-test for independent variables to test for mortalities and immune gene expression for interleukins, INF-I, INF-II, C3 and CD8. Mortality curve differences between groups were confirmed by comparing linear regression results applying Student’s independent *t*-test. Statistical differences in production variables such as the initial and harvest biomass, economic and biological food conversion rate and specific growth rate were determined using ANOVA. All analyses were carried out using the software Prism GraphPad^®^, version 9.3.1.

## 3. Results

### 3.1. In Vivo Efficacy of Quillaja Saponin Extracts against P. salmonis in Freshwater Conditions under Intraperitoneal Challenge

The general clinical appearance of the fish used in this study was acceptable and was within what is considered normal for the salmon industry, and therefore was representative of the Chilean salmon industry. The results shown in Figure 1 indicate that both quillaja extracts reduced mortality associated with *P. salmonis* infections. PAQ-Xtract showed a significatively higher survival rate than the NPQ group (70.9% vs. 62.5% survival, Student’s *t*-test = 17.7, df = 1, *p* = 0.03, Figure 1).

IFN-ΙΙ expression in fish kidneys and spleens was significantly higher for both quillaja extracts compared to the control group before the trial (Figure 2). The gene expression of IFN-II in surviving fish at the end of trial in their kidneys and spleens was increased two-fold for groups receiving quillaja extracts compared to control groups (Figure 2). However, IFN-II gene expression in the kidneys and spleens of fish that died during the trial was significantly lower than in control groups (fish infected and no quillaja extracts).

C3 expression in the kidneys and spleens can be seen in Figure 3. NPQ increased the C3 expression in all condition tested (prior to trial, in mortalities and survivors) compared to the control and PAQ-Xtract groups.

### 3.2. In Vivo Efficacy of Quillaja Extracts against P. salmonis in Atlantic Salmon Smolts, under a Cohabitation Challenge

Fish containing PAQ-Xtract in the feed showed a reduction in mortality of 16.8% and 30.3% in relation to the control groups that reached 60% and 90% mortality, respectively. Fish fed with NPQ in the feed showed similar mortalities to the Control group, which means that it did not reduce mortality at either mortality point (Figure 4).

Both quillaja extracts significantly increased the expression of both IFN-I and CD8 after 30 days of pre-trojan inoculation; however, after 15 days of inoculation, only CD8 was overexpressed. In fact, the level of IFN-1 in quillaja-treated fish 15 days after inoculation was lower than that in the control group without quillaja extract and infected with *P. salmonis* (Figure 5 and Figure 6, respectively).

The expression of interleukins 1, 8 and 12 is shown in Figure 7, Figure 8 and Figure 9. PAQ-Xtract induced a higher average expression of IL-1 after 15 days of pre-trojan inoculation; however, the average expression seems to be reduced at the end of the trial. These results must be carefully examined due to the relatively high standard deviation of the results at day 15. IL-8 expression was significantly higher in the group treated with NPQ and not PAQ-Xtract at the end of the trial, while IL-12 expression was not affected by neither of the extracts.

The expression of INF-II was significantly higher using both extracts at the end of the trial (Figure 10).

### 3.3. Efficacy of PAQ-Xtract in the Prevention of P. salmonis Infections in Atlantic Salmon in Seawater under Pilot Commercial Conditions

The field trial under pilot commercial conditions at the Huenquillahue center of MOWI Chile showed that the PAQ-Xtract groups displayed significatively reduced total mortality by the end of the trial by 20.6% compared to the control groups (PAQ-Xtract, 5.0% vs. control, 6.3% total mortality, F = 11.29, df = 1, *p* < 0.001). More specifically, PAQ-Xtract reduced mortality caused by *P. salmonis* by 71.3% compared to the control groups (PAQ-Xtract, 0.51% vs. control, 1.78% mortality by SRS, Student’s *t*-test = 6.35, df = 770, *p* < 0.001, Figure 11). Also, the PAQ-Xtract groups displayed higher biomass at the end of the trial compared to the control groups by 72.9% (283 kg vs. 1046 kg, t = 5.91, df = 109.5, *p* < 0.001).

There was an increase in production parameters in the PAQ-Xtract groups compared to the controls (Table 2), but this was only significant for the biological feed conversion rate (*p* < 0.005). Also, there was a reduction in the use of antibiotics in the PAQ-Xtract groups, with only one group needing antibiotic treatment due to *P. salmonis* infection, whereas all control groups required antibiotic administration (one cage vs. three cages, X^2^ = 3, df = 1, *p* = 0.08). Generally, antibiotics were administered orally through food for a period of 10 to 15 days at a time.

## 4. Discussion

The results obtained from this research are consistent with previous studies on the impact of quillaja extract in other fish species. For example, lower mortalities and the activation of cytokines have been reported in juvenile Pacu fish (*Piaractus mesopotamicus*) after seven days of supplementation with quillaja extracts and a subsequent challenge with the bacterium *Aeromonas hydrophila* [25]. A study in carp (*Cyprinus carpio*) also showed higher survival rates when challenged with *A. hydrophila*, as well as increased phagocytic and complement activity upon feeding with a diet supplemented with saponins [26]. In our study, we found higher survival rates in the PAQ-Xtract group compared to the NPQ and control groups. The differences observed in the impact of both quillaja extracts may be attributed to the higher saponin content (PAQ-Xtract has 2.5 times more saponins than NPQ). As described in the literature, saponins are directly involved in the activation of the immune system required to defend against intracellular pathogens [25]. These protective effects that we found could be associated with an early activation of INF-II and effector proteins such as the C3 complement factor and CD8+ T cells, inducing a protective microenvironment for the control of *P. salmonis*. These results with purified quillaja extracts (PAQ-Xtract) could be related to the direct activation of antigen-presenting cells (macrophages, dendritic cells, B cells) due to the ability of the quillaja extracts to integrate into the macrophage through cholesterol-dependent endocytosis, and as such accumulate in the membrane of the endosome and lysosome, destabilizing the membrane of these organelles, and subsequently activating Syk kinases, key signaling molecules in the activation of Nuclear Factor kappa β (NF-kβ) and the induction of a proinflammatory transcriptional program [27]. All this evidence taken together suggests that quillaja saponins can directly modulate cytokine expression in cells, irrespective of a pathogen infection. Moreover, the profile of expression of immune markers determined in this study could be correlated with the effect found in the commercial pilot trial, where during a natural outbreak of piscirickettsiosis, only 33.4% of the fish fed with PAQ-Xtract required antibiotic treatment to combat the disease. These results suggest that PAQ-Xtract was modulating the immune system when the outbreak of piscirickettsiosis reached the cages at around day 160, because mortalities never reached the threshold at which sanitary regulations triggers the use of antibiotics. It should be noted that this evaluation was carried out in pilot but industrial conditions, where different biotic and abiotic factors influence the daily and cumulative infectious mortality per week, obliging salmon producers to administer antibiotics or not. In the case of PAQ-Xtract during this period, this threshold was not reached. This situation was a substantial improvement compared to the control fish fed without quillaja extract, as after the disease outbreak, all of these fish required antibiotic treatment. These results suggest that antibiotic treatment could be significantly reduced in fish with a robust immune response (innate and cell adaptative) because of feeding with PAQ-Xtract. Future studies must be performed to validate this observed correlation, including the determination of the cytokine levels after the bacterial challenge, in order to confirm this hypothesis.

Cytokines are key players in the communication between innate immune cells, adaptive immune cells, and non-immune cells; their imbalance induced by *P. salmonis* contributes to creating an environment that favors its replication and establishment in the host [10]. Thus, the modulation of the levels of signal cytokines and effectors from the innate and adaptive immune response found in our study after the administration of quillaja saponins (such as IL-1β, IL-8, INF-I, INF-II, C3 and CD8 T cells) may help the host’s ability to reduce or stop an infectious process. In other vertebrate species such as mice, it has been demonstrated that the oral administration of quillaja saponin can change the profile of cytokines associated with allergies and the antigen-specific immune responses through the regulation of Th1/Th2 cytokine balance [28]. Other in vivo studies have shown that quillaja saponins can modulate the balance of cytokines, even in the absence of a pathogenic infection [10,29,30]. In our study, both quillaja extracts significantly increased the gene expression of C3, INF-I and INF-II in the kidney and spleen, before and after the challenge. This immune induction before the challenge may be regarded as a preparation stage of the cells of the innate and the adaptive immunity systems in order to mount a robust response against pathogen infection. The expression levels of cytokines and early induced effector molecules during an infection are an indicator of the ability of the host to reduce or stop the infectious process. The mechanisms that operate in this early response inhibit the replication of the pathogenic microorganism, or even destroy it. Thus, the modulation of primary immune signals such as IL-1β and IL-8, secondary immunological signals such as INF-I and INF-II, and effector molecules such as C3 from the innate immune system, as well as CD8+ T cells from the cellular mediated immune response, is an indicator of an active response produced by the host to inhibit intracellular replication of the pathogens in infected cells [7]. The saponin-triggered induction of key immunological markers (as INF-II, the complement effectors protein C3 and CD8+ T cells) in the absence of an infection in the control *P. salmonis* group, shows that quillaja extracts can support preparing cells through innate and adaptive immunity to combat pathogen challenges. Our results are further supported by previous reports showing that quillaja saponins can induce the direct activation of antigen-presenting cells (macrophages, dendritic cells), as well as expression of cytokines such as IL-1β, IL-6, IL-8, IL-18, IL-12, and TNF-α, even in the absence of pathogens [27]. Good macrophage activation, migration, phagocytic capacity, degradation of pathogens and presentation of foreign and self-antigens play key roles in bacterial infections [7,31,32]. Studies using the oral addition of quillaja extracts in yellowtail fish (*Seriola quinqueradiata*) showed an increase in macrophage mobility and phagocytic capacity and an enhanced immune response [33]. A similar result was obtained in humans and mice, where a purified quillaja saponin extract was delivered orally to evaluate its immune-stimulating potential [34,35,36,37]. Also, studies in crabs (*Portunus trituberculatus*) show that the oral addition of quillaja extracts significantly increases the phagocytic activity of macrophages and resistance to challenges by *Vibrio algynolithicus* [38]. Pathogen clearing without host damage depends on the balance between the production of different types of cytokines [39]. Some bacteria manipulate cytokines by releasing virulence factors through secretory mechanisms. As a result, macrophage activation is compromised, and IL-10 overexpression interferes with the innate immune system’s anticipated responses. *P. salmonis* can influence the actions of host cytokines and probably functions as a virulence factor that encourages intracellular bacterial reproduction in trout and salmon leukocyte cell lines. To promote microbial proliferation and ultimately survival within macrophages, *P. salmonis* may control the host cytokine response to upregulate IL-10-deactivating macrophages [10]. Future studies should be conducted measuring anti-inflammatory cytokines such as IL-4, IL-10, TGFβ, or transcription factor fox-p3 to determine the potential of purified quillaja saponins to regulate the balance between pro- and anti-inflammatory responses against *P. salmonis* in in vivo trials. It is important to take account of the fact that a dynamic balance of pro- and anti-inflammatory signals is the base for controlling disease and limiting potential pathologies [39]. A recent study demonstrated that rainbow trout splenocytes stimulated with Interferon and *P. salmonis* proteins demonstrated an up-regulated expression of the transcription factor fox-p3, which can induce a polarization towards a Treg phenotype [40], while a polarization to an anti-inflammatory milieu and a downregulation of the cellular mediated immunity response led by cytotoxic CD8+ T cells did not correlate with the protection sought with strategies such as vaccines against piscirickettsiosis under field conditions [7].

Depending on the plant source from which they are obtained, saponins have a variety of biological effects. Some studies showed that soya saponins can cause enteritis in fish [41,42], effects that can be related with a high daily inclusion rate given to the animals and the presence of other antinutritional factors in products like soybean meal (SBM). Other studies show saponins included in fish diets have economic benefits. Adding saponin supplements to the diets of tilapia and common carp caused increased growth in fish [43]. Similar results were seen in Nile tilapia over a 14-week feeding period at a higher dietary supplementation dose of 300 mg/kg of saponins [44]. Researchers discovered that adding saponins to diets promotes better feed energy retention and more effective protein utilization in fish and ruminants [44]. Although the compounds found in standard medicines known as saponins are familiar to scientific and industrial organizations, little was known about their mechanisms of action at the time, such as protein utilization, growth stimulation, fish reproduction or metabolic rates. A possible explanation for the overall beneficial effects of saponin may be its ability to alter the permeability of the intestinal membrane and its interference with the absorption of essential nutrients [43].

The used of PAQ-Xtract as a natural non-pharmacological product could be a potential strategy to reduce mortalities due to piscirickettsiosis, helping to reduce the current economic losses of salmon farming in Chile caused by this disease [45]. In addition, the use of natural extracts could contribute to the sustainability of production and reduce the use of antibiotics. The interest in these products emanates from their facile preparation protocols, economic viability, minimized ecological footprints and, as seen in the scientific evidence dedicated to the deployment of phytocompounds, their antimicrobial and antiviral capacity and their ability to increase both the innate and antigen-specific immunological response [46]. Nonetheless, the seamless integration of natural extracts into aquaculture hinges crucially upon surmounting the intricate challenge of instituting cost-effective and orally administered therapeutic phytocompounds [47].

In our three studied models, we showed that quillaja extracts with different levels of purification in saponin and non-saponin fractions added to the diet of Atlantic salmon before and after challenge with *P. salmonis* induce different impacts on animal survival and can lead to differences in the quantity and quality of the immunomarkers evaluated.

## 5. Conclusions

The dietary addition of PAQ-Xtract to Atlantic salmon modulates key markers in the innate and the adaptive cellular immune response which are critical to destroying cells infected with intracellular pathogens such as *P. salmonis*, increasing the fish survival rate under intraperitoneal, cohabitation and commercial pilot challenge conditions. Also, PAQ-Xtract allows us to significantly reduce mortalities and the use of antibiotics, positively affecting the productive variables in the pilot commercial conditions. The studies demonstrated that PAQ-Xtract is a potential natural non-pharmacological strategy for the prevention and control of *P. salmonis* infections in salmon.

## 6. Patents

The use of quillaja saponins extracts to prevent SRS was patented in Chile, United State of America, Canada and Norway under registration patent numbers 64.922, 10,987,393 B2, 2,972,175, and 345,485 B1, respectively.

## Figures and Tables

**Figure 1 animals-13-02845-f001:**
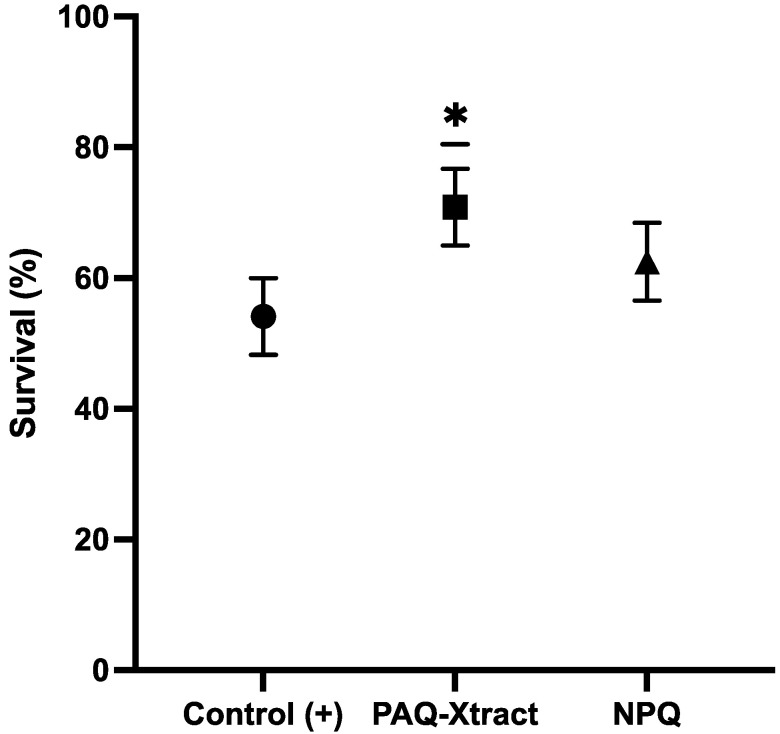
Survival rate of Atlantic salmon fry according to experimental group in a challenge with *P. salmonis* under the intraperitoneal infection model in freshwater (Control (+): fish infected and not treated; PAQ-Xtract: fish infected but treated with highly purified quillaja extract; NPQ: fish infected but treated with non-purified quillaja extract). * = statitistical significance *p* < 0.05.

**Figure 2 animals-13-02845-f002:**
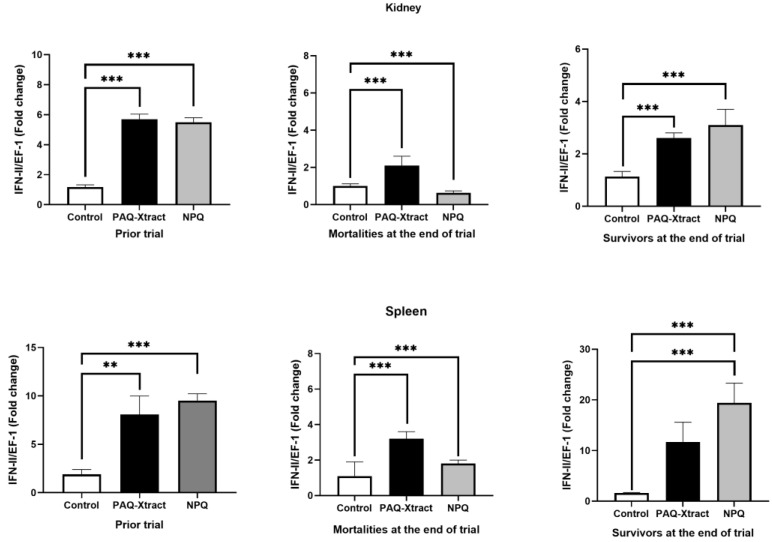
Expression of IFN-II in fish fed with two quillaja extracts, before (prior) and after challenge at the end of the freshwater trial. Bars represent the average and standard deviation of the fold change in the gene expression, with respect to the control group. Asterisks represent statistically significant differences according to the Mann–Whitney test (** *p* < 0.01, *** *p* < 0.001).

**Figure 3 animals-13-02845-f003:**
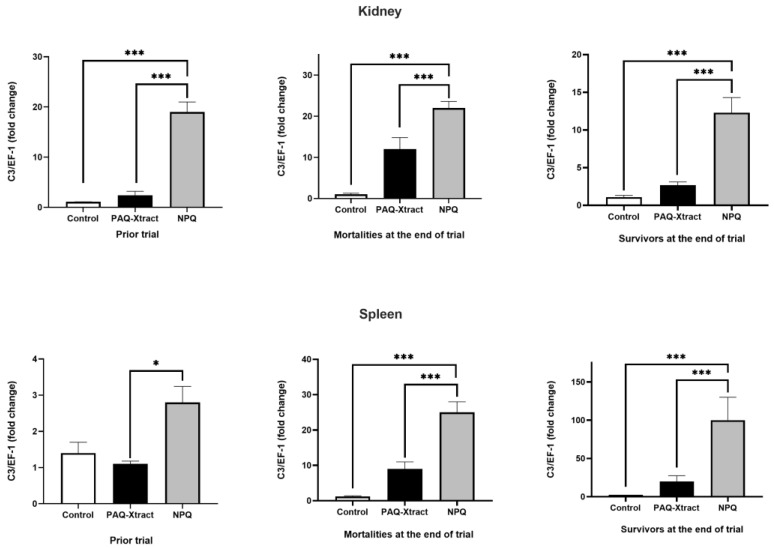
Expression of C3 in fish fed with two quillaja extracts, before (prior) and after challenge. Bars represent the average and standard deviation of the fold change in the gene expression, with respect to the control group. Asterisks represent statistically significant differences according to the Mann–Whitney test (* *p* < 0.05, *** *p* < 0.001).

**Figure 4 animals-13-02845-f004:**
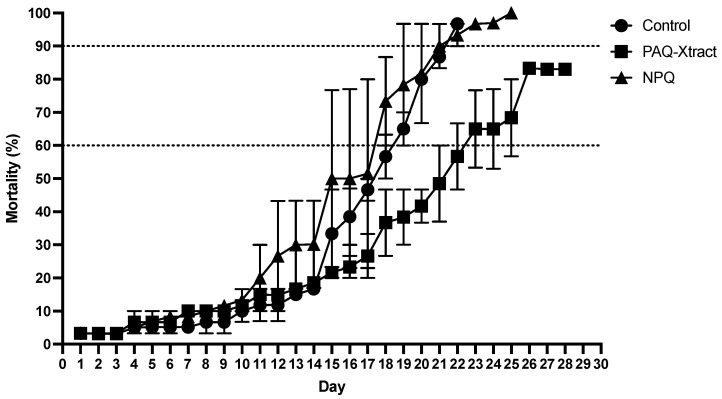
Mortality values of control and quillaja-saponin-extract-treated groups at the end of the challenge with *P. salmonis* under the cohabitation infection model.

**Figure 5 animals-13-02845-f005:**
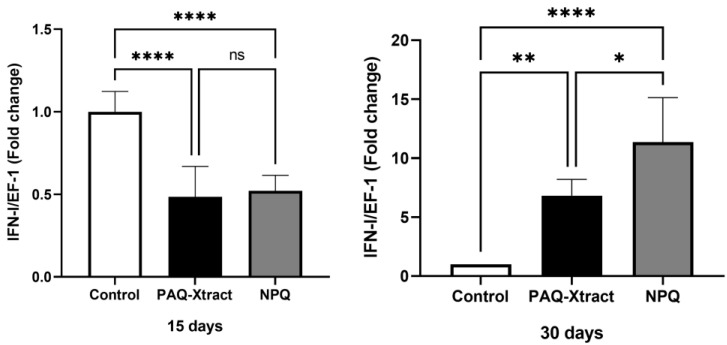
Expression of IFN-I in fish supplemented with two quillaja extracts at 15 and 30 days under the cohabitation infection model. Bars represent the average and standard deviation of the fold change in the gene expression, with respect to the control group. Asterisks represent statistically significant differences according to the Mann–Whitney test (* *p* < 0.05, ** *p* < 0.01, **** *p* < 0.0001).

**Figure 6 animals-13-02845-f006:**
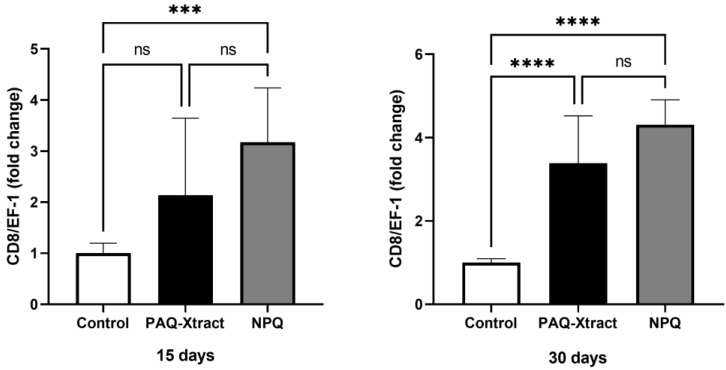
Expression of CD-8 in fish supplemented with two quillaja extracts at 15 and 30 days under the cohabitation infection model. Bars represent the average and standard deviation of the fold change in the gene expression, with respect to the control group. Asterisks represent statistically significant differences according to the Mann–Whitney test (*** *p* < 0.001, **** *p* < 0.0001).

**Figure 7 animals-13-02845-f007:**
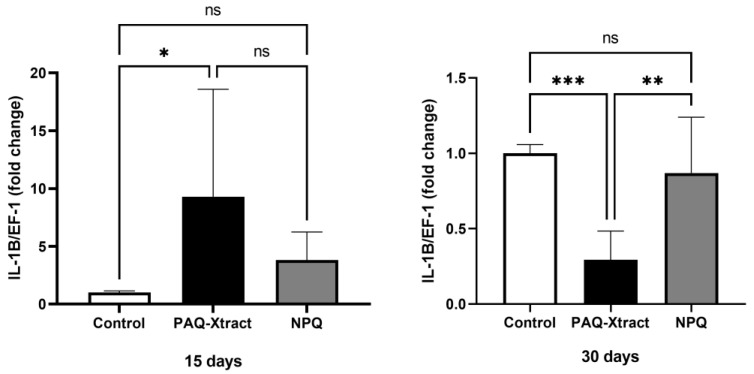
Expression of IL-1 in fish supplemented with two quillaja extracts at 15 and 30 days under the cohabitation infection model. Bars represent the average and standard deviation of the fold change in the gene expression, with respect to the control group. Asterisks represent statistically significant differences according to the Mann–Whitney test (* *p* < 0.05, ** *p* < 0.01, *** *p* < 0.001).

**Figure 8 animals-13-02845-f008:**
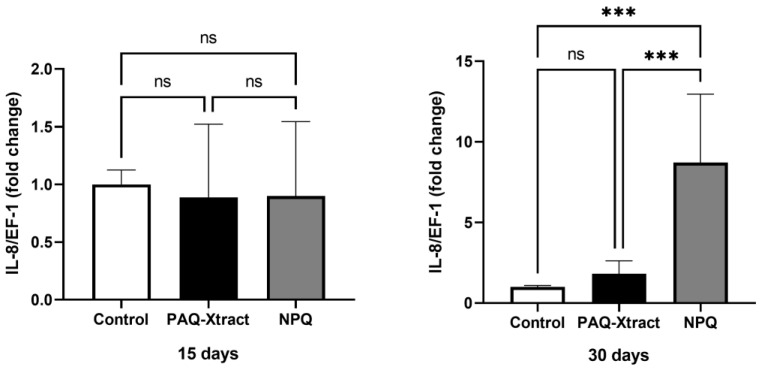
Expression of IL-8 in fish supplemented with two quillaja extracts at 15 and 30 days under the cohabitation infection model. Bars represent the average and standard deviation of the fold change in the gene expression, with respect to the control group. Asterisks represent statistically significant differences according to the Mann–Whitney test (*** *p* < 0.001).

**Figure 9 animals-13-02845-f009:**
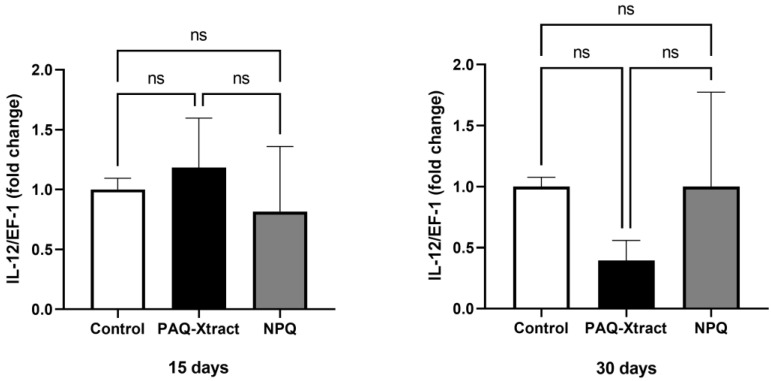
Expression of IL-12 in fish supplemented with two quillaja extracts at 15 and 30 days under the cohabitation infection model. Bars represent the average and standard deviation of the fold change in the gene expression, with respect to the control group.

**Figure 10 animals-13-02845-f010:**
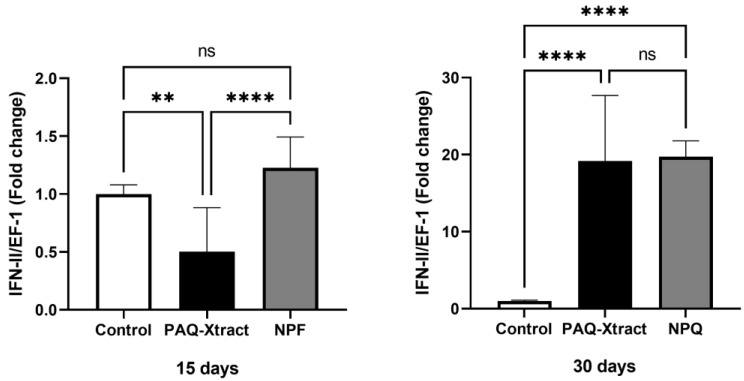
Expression of IFN-II in fish supplemented with two quillaja extracts at 15 and 30 days under the cohabitation infection model. Bars represent the average and standard deviation of the fold change in the gene expression, with respect to the control group. Asterisks represent statistically significant differences according to the Mann–Whitney test (** *p* < 0.01, **** *p* < 0.0001).

**Figure 11 animals-13-02845-f011:**
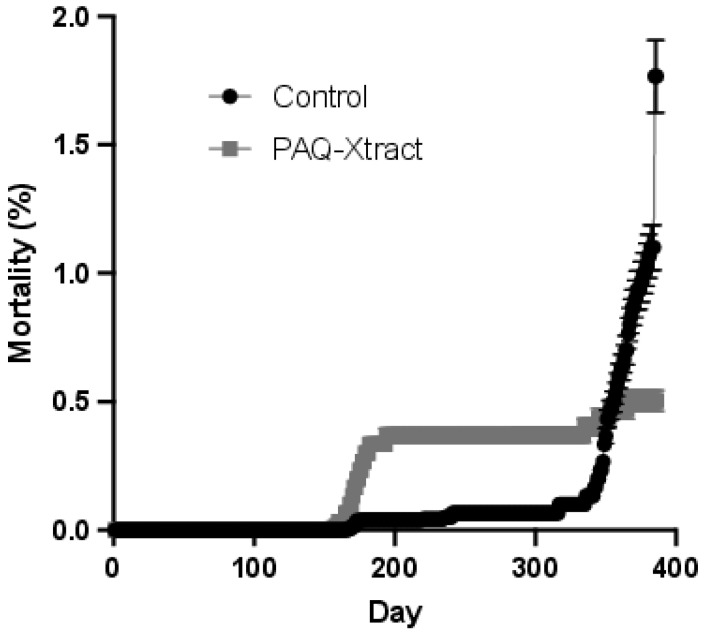
Mortality rate due to *P. salmonis* infections for the PAQ-Xtract and control groups during field trial.

**Table 1 animals-13-02845-t001:** Primer sequences for immune gene expression analysis.

Gene		Sequences (5′-3′)	Reference/ Accession Number GenBank
*IFN-I*	F R	CCTGCCATGAAACCTGAGAAGA TTTCCTGATGAGCTCCCATGC	NM_001123710.1
*IFN-II*	F R	TTCAGGAGACCCAGAAACACTAC TAATGAACTCGGACAGAGCCTTC	AY795563.1
*IL-8*	F R	GCAACAGCGGTCAGGAGATT TGGAATGATTCCCCTTCTTCA	HM162835.1
*IL-12*	F R	CTGAATGAGGTGGACTGGTATG ATCGTCCTGTTCCTCCG	XM_014205516.1
*cd8*	F R	CACTGAGAGAGACGGAAGACG TTCAAAAACCTGCCATAAAGC	AY693393.1
*IL-1β*	F R	CAAGCTGCCTCAGGGTCT CGGCACCCTTTAACCTCTCC	NM_001123582.1
*C3*	F R	TCCCTGGTGGTCACCAGTACAC ATGATGGTGGACTGTGTGGATC	[17]
*EF-1*	F R	CCCCTCCAGGACGTTTACAAA CACACGGCCCACAGGTACA	NM_001123629.1

**Table 2 animals-13-02845-t002:** Productive variables and antibiotic use of Atlantic salmon supplemented with PAQ-Xtract in a pilot scale trial under field conditions (Huenquillahue Center, MOWI Chile).

Group	Control	PAQ-Xtract
Initial biomass (kg)	1619 ± 258	1556 ± 410
Biomass at harvest (kg)	51,634 ± 6691	57,200 ± 4527
Economic feed conversion rate	1.18 ± 0.06	1.13 ± 0.04
Biological feed conversion rate	1.15 ± 0.02 ^a^	1.09 ± 0.03 ^b^
Specific Growth Rate	0.93 ± 0.01	0.94 ± 0.05
Antibiotic treatment	100.0%	33.3%

^a^ and ^b^ represent statistical differences at *p* < 0.005.

## Data Availability

The data presented in this study are available upon request from the corresponding author.

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
