# Peer review of "In Vivo Efficacy of Purified Quillaja Saponin Extracts in Protecting against Piscirickettsia salmonis Infections in Atlantic Salmon (Salmo salar)"

_animals, 2023, doi:10.3390/ani13182845_

Round 1
Reviewer 1 Report
It is a very interesting work about an important problem in salmonid aquaculture.
It could be improved:
1.- adding details in the description of powder extract compositions, especially non purified extract.
2.- description of the figures: Fig. 11, shows no standard deviation.
3.- definition about 100% antibiotic treatment.
MS is focused on the Quillay extract application to control P. salmonis infection, the topic is original. MS describes a new option to control P. salmonis.Author Response
Santiago, 9th August, 2023
Dear Reviewer 1
We appreciate the comments made, because improve the content of the manuscript. All comments were attended.
Reviewer #1
1.- adding details in the description of powder extract compositions, especially non purified extract.
Response:
Additional information of the composition of both products was added in “2.1. Quillaja saponin extracts”.
Lines 139-141: As described by FAO JECFA, Quillaja extract type II has saponins content of at least 65% and not more than 90% in solids, compared to Quillaja extract type I that has between 20 to 26% of saponins.
Additional information, but non included in the text:
Additionally to saponins, Quillaja extracts contain a non saponin fraction (NSF) including phenolic compounds, being (+)-piscidic acid the main phenolic component, and others in minor quantities like p-coumaroyl esters named as quillajaside A, and quillajaside B (Maier et al, 2015; Maier et al 2015).
Maier C, Conrad J, Carle R, Weiss J, Schweiggert RM. Phenolic constituents in commercial aqueous Quillaja (Quillaja saponaria Molina) wood extracts. J Agric Food Chem. 2015 Feb 18;63(6):1756-62. doi: 10.1021/jf506277p
Maier C, Conrad J, Steingass CB, Beifuss U, Carle R, Schweiggert RM. Quillajasides A and B: New Phenylpropanoid Sucrose Esters from the Inner Bark of Quillaja saponaria Molina. J Agric Food Chem. 2015 Oct 14;63(40):8905-11. doi: 10.1021/acs.jafc.5b03532
2.- description of the figures: Fig. 11, shows no standard deviation
Response:
Figure has been replaced with anew one showing standard deviations
3.- definition about 100% antibiotic treatment
Response:
The parameter, antibiotic treatment, was defined in materials and method.
Lines 241-243: Antibiotic treatment was measured as the percentage of cages that required the use of antibiotics at least once during the trial.

Reviewer 2 Report
The manuscript titled “In vivo efficacy of purified Quillaja saponin extracts for Piscirickettsia salmonis infections in Atlantic salmon (Salmo salar)” provided a promising method to fight against Piscirickettsia salmonis. The present study is general well-designed and conducted. Here are some minor comments:
1. A major concern of mine is that there were few typical immune parameters. The phenotypic immune response is a direct indicator of fish health status.
2. Line 77-85, it is mentioned that “When P. salmonis infects a host, it is phagocytosed by macrophages of the mu-77 cosa at the gills, skin, and gastrointestinal tract.” However, the evaluation of macrophage status was not involved in this study.
3. Line 147-153, please add information about the general profile of functional compounds in this PAQ-Xtract product. A general description must be added.
4. The ELF-1 gene was used as the reference gene. Did you have screening process for reference gene? There should be screening process to make sure that the ELF-1 gene is suitable reference gene.
5. Please add the primer sequence information.
6. Line 240, why was there a 1 month rest interval?
7. Explain NPQ and PAQ in the figure note. Figures should be independently readable.
8. Figure 2, if the data is expressed as fold change, one group should be normalized to be 1. However, it seems not the case here. For the panel “Spleen-mortalities at the end of the trial”, which group was normalized to be 1? Such things happened in other figures too. Please check them.
9. Figure, using different colors for different groups may make the Figure clearer and more readable.
10. Figure 11, between day 200 and 350, it seems that the treatment group performed worse than the control group.
11. The result and discussion parts are generally acceptable. A evaluation of the economics of this Quillaja extract methods could be added when applied in fish farming.
Author Response
Santiago, 9th August, 2023
Dear Reviewer 2
We appreciate the comments made, because improve the content of the manuscript. All comments were attended.
Reviewer #2
- A major concern of mine is that there were few typical immune parameters. The phenotypic immune response is a direct indicator of fish health status
Response:
We would have liked to quantify the gene expression of a greater number of markers of the innate and adaptive immune response, both cellular and humoral, to see if quillaja extracts could induce a more robust immune response, but there were economic and operational limitations that did not allow it. In any case, we seek to include markers such as Interferon type II and I, T CD8+, C3, which are some of the most affected in an active infection with P. salmonis.
In the case of the study with intraperitoneal (IP) challenge, evaluations were included before and after challenge, and determinations of "safety", "immunogenicity" and "efficacy in reducing mortalities by the pathogen", both in dead and surviving fish were preferred. This in the end forced us to do without some immune markers that better complemented the results.
Faced with this situation, we have continued to carry out research and validation under industrial conditions. To do this, we are monitoring innate immune markers (IL-12, IL-10, antimicrobial peptides such as cathelicidin), and adaptive T CD4+, T CD8+, Interferon gamma, and humoral (immunoglobulins IgT and IgM) markers in mucous membranes (gills, intestine) and systemic (kidney, spleen). These studies, which are ongoing in 2023, are making it possible to confirm that purified extracts of quillaja can help reduce mortalities, use of antibiotics, and modulate immune markers related to the control of intracellular pathogens such as P. salmonis, even under challenging field conditions.
- Line 77-85, it is mentioned that “When P. salmonis infects a host, it is phagocytosed by macrophages of the mucosa at the gills, skin, and gastrointestinal tract.” However, the evaluation of macrophage status was not involved in this study
Response:
In agreement with this comment, the paragraph related to infection process and virulence factor of P. salmonis was deleted.
“When P. salmonis infects a host, it is phagocytosed by macrophages of the mucosa at the gills, skin, and gastrointestinal tract. This bacterium can modulate the host immune response through their virulence factors such as Type IVB secretion systems [12,13] that allow exporting proteins and genetic material, to adhere, invade and replicate into intracellular medium of the host [14–16]. This process circumvents the host adaptive immune response required to destroy the infected cells”
- Line 147-153, please add information about the general profile of functional compounds in this PAQ-Xtract product. A general description must be added
Response:
Additional information of the composition of both products was added in “2.1. Quillaja saponin extracts”.
Lines 139-141: As described by FAO JECFA, Quillaja extract type II has saponins content of at least 65% and not more than 90% in solids, compared to Quillaja extract type I that has between 20 to 26% of saponins.
Additional information, but non included in the text:
Additionally to saponins, Quillaja extracts contain a non saponin fraction (NSF) including phenolic compounds, being (+)-piscidic acid the main phenolic component, and others in minor quantities like p-coumaroyl esters named as quillajaside A, and quillajaside B (Maier et al, 2015; Maier et al 2015).
Maier C, Conrad J, Carle R, Weiss J, Schweiggert RM. Phenolic constituents in commercial aqueous Quillaja (Quillaja saponaria Molina) wood extracts. J Agric Food Chem. 2015 Feb 18;63(6):1756-62. doi: 10.1021/jf506277p
Maier C, Conrad J, Steingass CB, Beifuss U, Carle R, Schweiggert RM. Quillajasides A and B: New Phenylpropanoid Sucrose Esters from the Inner Bark of Quillaja saponaria Molina. J Agric Food Chem. 2015 Oct 14;63(40):8905-11. doi: 10.1021/acs.jafc.5b03532
- The ELF-1 gene was used as the reference gene. Did you have screening process for reference gene? There should be screening process to make sure that the ELF-1 gene is suitable reference gene
Response:
As a control of the host cell, to normalize the qRT-PCR and the amount of RNA of each kinetic point, the Elongation Factor 1-Alpha (EF-1) gene was used, as recommended in previous studies with genes of reference (Peña et al, 2010).
Line 215-217: The expression of EF-1 was suggested as reference gene in RT-qPCR assays for studying the effect of P. salmonis on the host immune response (Peña et al, 2010).
Peña, A.A., Bols, N.C. & Marshall, S.H. An evaluation of potential reference genes for stability of expression in two salmonid cell lines after infection with either Piscirickettsia salmonis or IPNV. BMC Res Notes 3, 101 (2010). https://doi.org/10.1186/1756-0500-3-101
- Please add the primer sequence information
Response:
The primer sequence was incorporated in material and methods.
Line 218-226: The primer used in the study were the following: IFN-I (F: CCTGCCATGAAACCTGAGAA-GA; R: TTTCCTGATGAGCTCCCATGC), IFN-II (F: TTCAGGAGACCCAGAAACACTAC; R: TAATGAACTCGGACAGAGCCTTC), C3 (F : TCCCTGGTGGTCACCAGTACAC; R: ATGATGGTGGACTGTGTGGATC), IL-8 (F: GCAACAGCGGTCAGGAGATT; R: TGGAATGATTCCCCTTCTTCA), IL-12 (F: CTGAATGAGGTGGACTGGTATG; R: ATCGTCCTGTTCCTCCG), CD8 (F: CACTGAGAGAGACGGAAGACG; R: TTCAAAAAC-CTGCCATAAAGC), IL-1(F: CAAGCTGCCTCAGGGTCT; R: CGGCACCCTTTAACCTCTCC), EF-1 (F: CCCCTCCAGGACGTTTACAAA; R: CACACGGCCCACAGGTACA)
- Line 240, why was there a 1 month rest interval?
Response:
This strategy was defined based on the effect of PAQ-Xtract over fish immune system, were within a month of giving the product it was observed an activation of it, lasting for at least one more month.
According to the IP challenge here reported, Quillaja extracts can modulate key immunomarkers related with the control of intracellular bacteria, before and after challenge. This immuno modulation was observed in the fishes during the month evaluated, from the first week of the start giving Quillaja (prior challenge) and during the rest of the month. On the other hand, the “cohabitation challenge” demonstrated that the fishes giving Quillaja extracts, for one month before challenge, always modulate immuno markers related with protection.
It could be possible to give Quillaja extracts to the fish without negative nutritional and health effects. However, a cost-benefit relationship is always considered by salmon farmers, when introducing a health alternative within their tools to deal with health problems.
Previous studies using Quillaja extracts in salmon fishes (no reported here), demonstrated that Quillaja can be given into the diet of Atlantic salmon for more than three months, being safe for the animals. However, the real applications of this natural alternatives for the salmon producer, must be considered.
- Explain NPQ and PAQ in the figure note. Figures should be independently readable.
Response:
Notes have been added to figure notes.
- Figure 2, if the data is expressed as fold change, one group should be normalized to be 1. However, it seems not the case here. For the panel “Spleen-mortalities at the end of the trial”, which group was normalized to be 1? Such things happened in other figures too. Please check them.
Response:
Point considered. However, figures show the relation between the normalised gene EF-1 and each immune markers in each graph in the Y-axis. Therefore, there seems to be unnecessary to normalised something that is already done so.
- Figure, using different colors for different groups may make the Figure clearer and more readable
Response:
Point taken, however, instead of changing into colours, we used, white, black and gray to clearly identify the three groups. We modified the figures consistently in the manuscript. Hopefully, this will clarify each group.
- Figure 11, between day 200 and 350, it seems that the treatment group performed worse than the control group
Response:
The biological behavior at the beginning of these trials (days 200 and 350) could be explained because in the treatment group a percentage of salmon were susceptible to infection, but then the immunomodulation by quillaja extracts prevented the propagation of the bacteria and the percentage of deaths. On the other hand, in the control group, although the deaths were lower at the beginning, they always increased throughout the trial. Until a certain point was reached, the mortality rate could not be controlled except with antibiotic treatment.
Lines 399-405: These results suggest that PAQ-Xtract was modulating the immune system when the outbreak of SRS reached the cages around day 160, because mortalities never reached the threshold that sanitary determines the use of antibiotics. It should be noted that this evaluation was carried out in pilot but industrial conditions, where different biotic and abiotic factors influence the daily and cumulative infectious mortality (SRS) per week, which obliges to salmon producers to administer antibiotics or not. In the case of PAQ-Xtract during this period, this threshold was not reached.
- The result and discussion parts are generally acceptable. A evaluation of the economics of this Quillaja extract methods could be added when applied in fish farming
Response:
In agreement with this comment a following paragraph was included:
Lines 485-495: “The used of PAQ-Xtract as a natural non-pharmacological product could be a potential strategy to reduce mortalities due to SRS, helping reduce the current economic losses of the salmon farming in Chile caused by this disease (Henríquez, P. et al. 2016)”. In addition, the use of natural extracts could contribute to the sustainability of production and reduce the use of antibiotics. The interest in these products emanates from their facile preparation protocols, economic viability, minimized ecological footprints and, in the scientific evidence dedicated to the deployment of phytocompound elucidating their antimicrobial, antiviral capacity, and their ability to increase both the innate and antigen-specific immunological response (Bondad-Reantaso et al, 2023). Nonetheless, the seamless integration of natural extracts into aquaculture hinges crucially upon surmounting the intricate challenge of instituting cost-effective and orally administered therapeutic phytocompounds (Nik et al, 2022).”
Henríquez, P., Kaiser, M., Bohle, H., Bustos, P. and Mancilla, M. (2016), Comprehensive antibiotic susceptibility profiling of Chilean Piscirickettsia salmonis field isolates. J Fish Dis, 39: 441-448. https://doi.org/10.1111/jfd.12427
Bondad-Reantaso, MG, MacKinnon, B, Karunasagar, I, et al. Rev Aquac. 2023; 1-31. doi:10.1111/raq.12786
Nik Mohamad Nek Rahimi N, Natrah I, Loh JY, Ervin Ranzil FK, Gina M, Lim SE, Lai KS, Chong CM. Phytocompounds as an Alternative Antimicrobial Approach in Aquaculture. Antibiotics (Basel). 2022 Mar 31;11(4):469. doi: 10.3390/antibiotics11040469

Reviewer 3 Report
The manuscript presented to me for review is absolutely interesting and important from the global trends related to human nutrition point of view, especially in the context of aquaculture. The aspect of the possibility of limiting the antimicrobials using is also important here. Some ports of the manuscript, however, are far too long. Some require elaboration while others need to be supplemented. Detailed notes on the text are provided below.

Author Response
Santiago, 9th August, 2023
Dear Reviewer 3
We appreciate the comments made, because improve the content of the manuscript. All comments were attended.
Reviewer #3:
- 1. Line 45. Suggestion to add "saponins" to keywords.
Response:
In agreement with the reviewer comment, saponins was added as a keyword (Line 49).
- The absolute necessity to shorten the "Introduction" part, especially the fragment
between lines 74 to 133. Suggestion of shorter sentences and less detail of data. This
should be a general introduction to the subject of the article.
Response:
The introduction was shortened.
Specifically, we deleted information related with infection process and virulence factor from P. salmonis (“When P. salmonis infects a host, it is phagocytosed by macrophages of the mucosa at the gills, skin, and gastrointestinal tract. This bacterium can modulate the host immune response through their virulence factors such as Type IVB secretion systems [12,13] that allow exporting proteins and genetic material, to adhere, invade and replicate into intracellular medium of the host [14–16]. This process circumvents the host adaptive immune response required to destroy the infected cells”). Also, we summarized the main functions of cytokines. On the other hand, we believe that information about Quillaja Tree and Quillaja saponins is important in the context of this research.
- The need to specify in a separate paragraph at the end of the "Introduction" part the aim of the work, starting with the words: The aim of the study was ... The text from line 134 to the beginning of line 136 can be used literally here.
Response:
The aim of the work was specified in a separated paragraph.
Lines 136-143 were deleted (“To this end we tested two types of Quillaja extracts, orally administered to salmon un-der three in vivo conditions - two controlled and one at a pilot scale in commercial conditions. The results were employed to evaluate: i) the potential reduction in mortalities associated with P. salmonis infections; ii) the feasibility of reducing antibiotic use in the production of salmon fish, and iii) the impact on the modulation of some immune markers in fry and smolts of Atlantic salmon upon oral administration of the Quillaja extracts”)
- Lines 136-142. The need to move the content of this part of the manuscript to the chapter "Materials and Methods".
Response:
Lines 136-142 were deleted from the manuscript.
- Line 157. The need to provide the average weight ± SD of a single individual value.
Response:
Standard Deviation was added in the text.
Line 146: “One hundred and eighty (180) Atlantic salmon fry weighting 30 ± 1.4 g/each were used”
- Lines 163-165. This part has the results elements and should therefore be moved from this part of the manuscript or deleted.
Response:
In agreement with the reviewer comments, the sentences were deleted from material and methods and included in results.
Lines 255-257: “The general clinical appearance of the fishes used in this study was acceptable and was within what is considered normal for the salmon industry and therefore representative of the Chilean salmon industry”
- Line 166. How were daily food rations for fish determined? Also, regarding the whole experiment, what was the basic composition of the feed given to the fish.
Response:
We explained the required information, but it was not added to the manuscript.
To give the daily food ration, the fish were hand fed twice a day with the feed control and the feed with the Quillaja extracts, in a quantity determined according to the live weight, the expected growth rate of the fish and the water temperature. To be able to do this, the total fish of the IP and cohabitation trials were weighed on days 0, 7, 15, 30 and 60 after the beginning of the experiment, according to the trial, to obtain data of weight on each occasion. For the pilot trial, the feed was delivered at 100% of the feed requirement and adjusted weekly according to expected weight gain and temperature.
The optimal dose of Quillaja extracts used into the feed in these studies came from previous determinations (no reported here) where incremental doses of Quillaja extracts where given orally to the fish evaluating a range from 0.9 to 12.0 mg of saponins/Kg body weight of fish.
Feed composition:
The feed was obtained from the company Ewos Chile, but we were unable to access the exact composition of ingredients used. However, the basic composition of the feed used for the fish are:
- Vegetable protein: 30% (Soy flour; soy protein concentrated; sunflower; corn gluten meal; wheat gluten)
- Animal by-products: 20% (Poultry meal; Pork meal; Feather flour; Blood meals poultry oil; flour from entrails)
- Fish products: 15% (fish meal; fish flour; fish oil; other sub-products of fishing industry)
- Wheat: 15%
- Canola oil: 10%
- Water: 10%
The centesimal Analysis Food of the diet Ewos micro, was:
- Moisture: 9.0%
- Total Crude Protein: 50.0%
- Fat: 22.0%
- Crude fiber: 1.0%
- Ash: 9.5%
- Nitrogen-Free Extract: 8.5%
(Gross Energy: 22 MJ/kg DM)
- What was the number of repetitions in each variant of the experiment
Response:
In vivo efficacy of Quillaja saponin extracts against P. salmonis in freshwater conditions under intraperitoneal challenge were performed in duplicate.
In vivo efficacy of Quillaja saponin extracts against P. salmonis in Atlantic salmon smolts, under a cohabitation challenge were performed in duplicate.
Efficacy of PAQ-Xtract in the prevention of P. salmonis infections in Atlantic salmon at sea water under pilot commercial conditions were performed in triplicate.
- Lines 368 - 374. This part of the "Discussion" chapter repeats the elements of the "Results" part. This fragment should be modified or deleted, starting the chapter "Discussion" on line 375
Response:
Lines 368-374 were deleted.
- At the end of the "Discussion" section, a suggestion to add at least one paragraph and 2- 3 references about the possibility of reducing the use of antimicrobials in the context of the research conducted
Response:
In agreement with the reviewer comment, we included a following paragraph:
Lines 485-495: “The use of PAQ-Xtract as a natural non-pharmacological product could be a potential strategy to reduce mortalities due to SRS, helping reduce the current economic losses of the salmon farming in Chile caused by this disease (Henríquez, P. et al. 2016). In addition, the use of natural extracts could contribute to the sustainability of production and reduce the use of antibiotics. The interest in these products emanates from their facile preparation protocols, economic viability, minimized ecological footprints and, in the scientific evidence dedicated to the deployment of phytocompound elucidating their antimicrobial, antiviral capacity, and their ability to increase both the innate and antigen-specific immunological response (Bondad-Reantaso et al, 2023). Nonetheless, the seamless integration of natural extracts into aquaculture hinges crucially upon surmounting the intricate challenge of instituting cost-effective and orally administered therapeutic phytocompounds (Nik et al, 2022).”
Henríquez, P., Kaiser, M., Bohle, H., Bustos, P. and Mancilla, M. (2016), Comprehensive antibiotic susceptibility profiling of Chilean Piscirickettsia salmonis field isolates. J Fish Dis, 39: 441-448. https://doi.org/10.1111/jfd.12427
Bondad-Reantaso, MG, MacKinnon, B, Karunasagar, I, et al. Rev Aquac. 2023; 1-31. doi:10.1111/raq.12786
Nik Mohamad Nek Rahimi N, Natrah I, Loh JY, Ervin Ranzil FK, Gina M, Lim SE, Lai KS, Chong CM. Phytocompounds as an Alternative Antimicrobial Approach in Aquaculture. Antibiotics (Basel). 2022 Mar 31;11(4):469. doi: 10.3390/antibiotics11040469)
- Suggestion to shorten the "Conclusions" part to the one paragraph.
Response:
The Conclusion was shortened.
Lines 501-508: “The dietary addition of PAQ-Xtract to Atlantic salmons, modulate key markers from the innate and the adaptive cellular immune response which are critical to destroy cells infected with intracellular pathogens as P. salmonis, increasing the fish survival rate under intraperitoneal, cohabitation and commercial pilot challenge conditions. Also, PAQ-Xtract allows to significantly reduce mortalities and the use of antibiotics, positively affecting the productive variables in the pilot commercial conditions. The studies demonstrated that PAQ-Xtract as a potential natural non-pharmacological strategy for the prevention and control P. salmonis infections in salmon.”
- The need to check the correspondence of the reference cited in the text with its list at the end of the manuscript
Response:
All changes were highlighted in the manuscript (including the new references incorporated)
